# Predicting Egg Passage Adaptations to Design Better Vaccines for the H3N2 Influenza Virus

**DOI:** 10.3390/v14092065

**Published:** 2022-09-17

**Authors:** Yunsong Liu, Hui Chen, Wenyuan Duan, Xinyi Zhang, Xionglei He, Rasmus Nielsen, Liang Ma, Weiwei Zhai

**Affiliations:** 1Key Laboratory of Zoological Systematics and Evolution, Institute of Zoology, Chinese Academy of Sciences, Beijing 100101, China; 2University of the Chinese Academy of Sciences, Beijing 100049, China; 3Human Genetics, Genome Institute of Singapore, Agency for Science, Technology and Research, Singapore 138672, Singapore; 4MOE Key Laboratory of Gene Function and Regulation, State Key Laboratory of Biocontrol, School of Life Sciences, Sun Yat-sen University, Guangzhou 510275, China; 5Department of Integrative Biology, University of California-Berkeley, Berkeley, CA 94707, USA; 6Department of Statistics, University of California-Berkeley, Berkeley, CA 94707, USA; 7Globe Institute, University of Copenhagen, 1350 København, Copenhagen, Denmark; 8Center for Excellence in Animal Evolution and Genetics, Chinese Academy of Sciences, Kunming 650223, China

**Keywords:** H3N2 influenza, passage adaptation, vaccine efficacy, epistasis, fitness landscape, convergent evolution

## Abstract

Seasonal H3N2 influenza evolves rapidly, leading to an extremely poor vaccine efficacy. Substitutions employed during vaccine production using embryonated eggs (i.e., egg passage adaptation) contribute to the poor vaccine efficacy (VE), but the evolutionary mechanism remains elusive. Using an unprecedented number of hemagglutinin sequences (*n* = 89,853), we found that the fitness landscape of passage adaptation is dominated by pervasive epistasis between two leading residues (186 and 194) and multiple other positions. Convergent evolutionary paths driven by strong epistasis explain most of the variation in VE, which has resulted in extremely poor vaccines for the past decade. Leveraging the unique fitness landscape, we developed a novel machine learning model that can predict egg passage substitutions for any candidate vaccine strain before the passage experiment, providing a unique opportunity for the selection of optimal vaccine viruses. Our study presents one of the most comprehensive characterizations of the fitness landscape of a virus and demonstrates that evolutionary trajectories can be harnessed for improved influenza vaccines.

## 1. Introduction

As a dominant seasonal RNA virus, influenza infects 5–15% of the total population, leading to an annual mortality rate of more than half a million [1,2]. One of the major approaches used for preventing influenza infection is vaccination [3]. Due to the virus’ rapid evolution (i.e., antigenic drift) [4,5], the World Health Organization (WHO) organizes two consultation meetings every year and attempts to minimize the mismatch between the vaccine strains and circulating strains by selecting dominant circulating influenza strains as the vaccine strains. Despite many years of effort, vaccine efficacy (VE) against influenza viruses, especially the H3N2 subtype, remains extremely poor [6,7]. For example, the VE for the H3N2 type showed two rapid declines from an intermediate efficacy of 40–50% to less than 10% in the years 2011–2015 as well as the years 2016–2019 [8,9,10,11,12,13,14,15]. Understanding the viral and host factors driving this low vaccine efficacy remains a pressing task for the field [16].

Since the 1940s, a conventional step in vaccine production has been the growing of vaccine strains in embryonated eggs before mass production [17]. One of the key features distinguishing the egg and mammalian environments is the cellular receptor (i.e., sialylated glycans) in different species, with alpha-2,3 glycosidic bonds (SAa-2,3Gal) found in birds and alpha-2,6 glycosidic bonds (SAa-2,6Gal) in humans [18,19]. Due to differences in the cellular environments, growing human adapted influenza viruses in embryonated eggs leads to a strong positive selection, a process known as egg passage adaptation [19,20]. Many important residues in hemagglutinin (HA), such as 186 [21,22,23,24], 194 [25,26,27] and 219 [28], mutate frequently during egg passage. Since residues in the receptor binding sites (RBS) of hemagglutinin overlap significantly with the antigenic sites (i.e., antigenic sites A–E) [29,30], a significant proportion of egg passage substitutions in HA, such as H156Q [31,32], L194P [25,27] and T160K [33], have been shown to greatly impact viral antigenicity and, subsequently, vaccine efficacy. As the H3N2 influenza adapts to the human host, which has been the case since its zoonotic transfer from birds to humans in 1968 [34], the extent of the egg passage adaptation in H3N2 influenza has continuously increased [35], leading to extremely poor vaccine efficacy over multiple years [36].

In addition to sequence analysis, deep mutational scanning experiments examining the joint effects of multiple key residues in several H3N2 strains revealed a dynamically evolving fitness landscape [37,38,39], suggesting that the effects of individual substitutions in hemagglutinin depend on the genetic background of the strain. In the case of egg passage adaptation, a recent structural and functional study discovered that two dominant egg passage substitutions, G186V and L194P, are mutually exclusive, driven by their disruptive effects on the binding pocket [40]. These findings suggest that egg passage substitutions are often non-independent and might depend on the genetic background of the focal strain. Even though many sequencing and functional studies have been carried out in order to characterize individual substitutions in different egg-passaged strains [21,22,23,24,25,26,27,28,31,32,33], the joint effects of multiple egg passage mutations and how they might affect vaccine efficacy are still poorly understood.

In this study, using an unprecedent number of HA sequences (*n* = 89,853) sequences from the GISAID database [41] and a probabilistic approach known as mutational mapping [42,43], we explored the temporal dynamics of egg passage adaptation, focusing, in particular, on how substitutions across multiple egg passage residues might act cooperatively to drive the fluctuations in VE that have been observed over the past decade. We found that the fitness landscape of egg passage adaptation is governed by pervasive epistatic interactions between major egg-passage residues and the adaptive walks are constrained to only a few convergent evolutionary paths. Through machine learning approaches, we showed that the evolutionary trajectories of egg passage adaptation can be predicted precisely for any candidate vaccine virus ahead of the passage experiment, providing a unique means of selecting optimal strains for vaccine production. 

## 2. Results

### 2.1. Data Curation and Passage Annotation

From the GISAID database, we obtained all the influenza H3N2 hemagglutinin HA1 sequences recorded since 1968 (*n* = 96,745), including 58% of sequences collected after the year 2016. After multiple QC steps (Methods), 89,853 high-quality sequences were retained for subsequent analysis (denoted as dataset D1, Methods). Since there were multiple identical sequences in the dataset, we further created a non-redundant dataset of unique sequences (*n* = 37,938, denoted as dataset D2) to reduce the computational burden for further analyses. We used D2 for most of the analyses, unless stated otherwise. In addition to collecting the sequences, we also curated passage histories of all the sequences [44] and found that most of the sequences were grown in cell lines (e.g., MDCK or SIAT, Figure 1A) and only a small fraction of sequences were passaged in embryonated eggs (*n* = 716 in D2, Methods). 

### 2.2. Strong Positive Selection in the Key Residues Drives Egg Passage Adaptation

After multiple sequence alignment, we inferred the phylogenetic relationships of all the sequences [45] (Figure 1B, Methods). To capture the egg passage mutations, we focused on substitutions along the terminal branches of the egg sequences (Figure 1B). In addition to standalone terminals (denoted as egg terminals), multiple egg-passaged sequences can sometimes form a monophyletic clade (denoted as the egg clade, Figure 1B), and convergent mutations observed in multiple strains can be “mapped” (i.e., inferred) to internal branches, even though egg passaging and, hence, egg passaging adaptation likely occurred independently in each of our lab experiments. We thus defined branches of interests (denoted as egg branches) as all the branches in the egg clades, as well as egg terminals (i.e., red branches in Figure 1B). 

Given the observed sequences at the leaf nodes of a phylogeny, the mutational mapping method is a stochastic sampling procedure used to sample possible mutational histories of the sequences [42]. It provides one of the most efficient approaches for exploring patterns of adaptive evolution over very large phylogenies (e.g., *n* > 1000) [42,43,46]. The mutational histories include both the ancestral states of all internal nodes and changes along each branch. Using maximum likelihood estimates of the phylogenetic tree and mutational parameters (e.g., the GTR model and rate parameters across sites, Methods), these sampled evolutionary histories form the basis for exploring the temporal dynamics of the adaptive change. The advantage of this method is that it can model uncertainties regarding inferred ancestral states while incorporating complex mutational models [42,46]. 

Based on the sample histories derived from the mutational mapping method, we employed three statistical tests to identify the amino acid positions driving the egg passage adaptation. Particularly, when we first examined whether more nonsynonymous mutations occurred than expected by chance relative to the rate of synonymous mutations (the positive selection test), we identified 18 residues strongly driving the egg passage adaptation (Figure 1C, Methods). The intensity of the positive selection across these 18 residues is extremely strong, with an overall A(nonsyn)/S(syn) ratio as high as 31.72 (e.g., 793 nonsyn and 25 syn for a randomly sampled history). In addition, we found that egg passage substitutions mainly occur in two important residues, 186 and 194, with many other residues, such as 156, 219 and 225, also experiencing frequent mutations (Figure 1C). Even though most of the residues were in our previous studies [35,36], two new residues, 160 and 225, are identified in this work. Inspecting the properties of these 18 residues, we found that most of them overlap with known functional domains, such as the receptor binding sites (RBS) and antigenic B and D domains (Figure 1C), implying that receptor binding or host factors might drive egg passage adaptation. 

We further tested whether a given residue tends to have more mutations along the egg branches than expected by chance (the enrichment test) and whether there are convergent mutations at a given residue along egg branches (the convergent test, Methods). Many residues showed significant results in both tests, indicating that adaptive mutations are enriched in egg branches and tend to be highly convergent mutations. Taken together, our findings suggest that the egg passage adaptation was driven by strong positive selection at a set of key residues. 

### 2.3. Strong Temporal Fluctuations in Egg Passage Adaptation

The sample histories derived from the mutational mapping method provide a unique means of examining the temporal trajectories of egg passage substitutions in different residues. When we estimated the rate of mutation of the egg passage, we found that the substitution rate can be as high as 0.00144/site/passage (Methods). Using the frequency of substitution of each residue in different time windows along the egg branches (Methods), we found that several residues, including 186, 194 and 219, have rapidly increasing frequencies of substitution, suggesting that the intensity of positive selection at these positions has become significantly stronger in recent years (Figure 1D). Interestingly, the substitution frequencies of the two newly identified residues, 160 and 225, have been elevated greatly since the year 2000, in accordance with their recent appearance in egg passage adaptation (Figure 1D). When breaking down the overall substitution at a given residue into individual mutation types at the amino acid level, we found that different positions have distinct mutational profiles that change over the years. For example, residue 194 shifted from showing multiple types of changes (e.g., L194I or L194P) in its history to showing only the L194P substitution in recent years (Figure 1E). On the contrary, residue 225 has multiple substitution types that have occurred in recent years (Figure 1F). In summary, egg passage adaptation has a dynamic history of substitution, with strong temporal fluctuations over time.

### 2.4. Pervasive Epistatic Interactions Driving Egg Passage Adaptation

By inspecting the temporal dynamics of substitutions in different residues, we observed multiple residues with similar trends. For example, substitution frequencies at two dominant residues, 186 and 194, are both monotonically increasing, and two newly identified residues, 160 and 225, simultaneously increased around the year 2000 (Figure 1D). When we clustered the temporal trends of the most frequent (amino acid) substitutions (e.g., G186V and L194P), we found that multiple residues tend to have concerted evolutionary trajectories (Figure 2A), suggesting potential epistatic interactions driving the egg passage adaptation. We noted that the mutations tabulated here are occurring de novo in different phylogenetic lineages, and the concerted evolutionary trajectories are, therefore, not a consequence of linkage. Since the mutational mapping method allows us to infer the history of the mutations along all the egg branches, we tested whether substitutions in two residues tend to co-occur along the same egg branch (Figure 2B). For example, when the L194P substitution mutates on a branch, we found that D225G, T160K, and T203I tend to co-occur together (Figure 2C). In contrast, the mutation G186V does not seem to co-occur with L194P, despite having a similar temporal trend (Figure 2A). This suggests that similar temporal trajectories might not imply a positive correlation. In order to systematically investigate whether substitutions at two residues tend to occur more (i.e., co-occurrence) or less frequently (i.e., mutually exclusive) than expected by chance, we applied the contingency test and identified multiple positive (*n* = 16) and negative (*n* = 6) interactions between all the pairs among the 18 residues (Methods). For example, the two dominant residues 186 and 194, despite having similar temporal trends, tend to occur in a mutually exclusive manner along the egg branches (Figure 2D and Appendix A, *p*-value < 10^−6^) [40]. More strikingly, all the negative interactions are between these two dominant residues and other positions. In order to visualize the epistatic relationship, we plotted the epistatic network of these 18 residues. Interestingly, residues 186 and 194 each form a subnetwork with positive interactions within each subnetwork but pervasive negative interactions between subnetworks (Figure 2E). The two newly identified residues, 160 and 225, positively interact with the two dominant residues, as well as multiple residues from the two subnetworks, which might have enabled them to mutate very frequently in recent years. In summary, we found that egg passage adaptation is driven by concerted changes across multiple residues, resulting in a rich epistatic network headed by two dominant residues, 186 and 194. The highly epistatic nature of egg passage substitutions revealed a multimodal fitness landscape, where substitutions at dominant residues drive convergent mutational paths during passage adaptation (see later sections). 

### 2.5. Passage Adaptation Drives Low Vaccine Efficacy, as Observed Consistently over the Years

We previously demonstrated that there was a strong negative correlation between the strength of egg passage adaptation and vaccine efficacy (VE) during years 2010–2015 and that 75% of the variation in VE could be explained by egg passage adaptation [36]. As the adaptation was found to have strong temporal fluctuations (Figure 1D), we wondered how the temporal dynamics of adaptive evolution might affect vaccine efficacy over the years and whether the recent drop in vaccine efficacy over 2016–2019 can be attributed to the effects of egg passage adaptation. 

One important observation worth emphasizing is that the extent of positive evolution in the egg medium is extremely strong (i.e., often having zero or only a few synonymous changes, Figure 1C). Thus, the traditional metric used for measuring the extent of positive selection (i.e., *d_N_*/*d_S_*) is often exceedingly large and makes it difficult to capture temporal fluctuations in the extent of adaptive evolution. We thus developed multiple statistics measuring different aspects of this extreme form of positive selection. With the updated list of positively selected residues, as well as a greater number of sequences relative to our previous study [36], we first calculated a previously developed statistic known as the enrichment score (ES) [36]. The ES was defined as the ratio of the frequency of the allele (amino acid) in the egg strains (p_egg_) to the frequency of the allele in all the strains (p_all_) (i.e., p_egg_/p_all_, calculated for the full dataset D1, Methods). Highly adaptive alleles that are strongly selected in embryonated eggs tend to be enriched in egg-passaged isolates, while being uncommon in the non-egg strains, resulting in a high ES. Therefore, the ES is a useful measure of the relative selective advantage of a particular allele in egg passage conditions. Plotting the ESs of all alleles in the 18 residues, we found that there are many residues with very high enrichment scores, indicating a strong convergent or parallel evolution of increasing frequencies of these alleles in the egg-passaged strains (Appendix A). 

For a given sequence in the database, the enrichment scores of the alleles at the 18 residues can be used to define an 18-dimensional vector for that sequence. When we reduced the 18-dimensional vector for all 89,853 sequences down to two dimensions using principal component analysis (PCA), we found that most of the sequences cluster around a point (denoted as C(x_0_,y_0_)) close to the origin, indicating no signal of egg passage adaptation for most of the sequences (Appendix A, Methods). However, sequences deviating from the origin were grouped in discrete islands and were mostly egg-passaged strains (Appendix A inset). Thus, we previously defined the adaptive distance (AD) of a given strain [36] as the distance between the coordinates of that strain and C(x_0_, y_0_). The AD correlates with the level of egg passage adaptation and provides a unique metric for integrating the signal of adaptation across all the residues. In our previous study, we found a strong negative correlation between the AD and VE [36]. With the larger dataset, we found a similar landscape for the AD, and the two leading PCs were dominated by residues 186 and 194, respectively (Appendix A). 

Interestingly, when we plotted the proportions of different alleles in the egg-passaged sequences of the vaccine strains over the years (there are multiple egg-passaged sequences for a given vaccine strain in the database, see Appendix A), we observed two sharp increases in the frequency of allele V at residue 186 over years 2013–2016 and allele P at residue 194 over years 2016–2019 exactly matching two rapid declines in the VE, suggesting that these two residues, together with their positively interacting changes, may drive the strong temporal fluctuations in the VE (Appendix A). It is interesting to observe that the short recovery of the vaccine efficacy in 2016 seems to have occurred at the time junction where the vaccine strains’ mutational preference changed from G186V to L194P, driven by strong negative epistasis between the two residues. Taken together, these findings indicate that passage adaptation has driven the low vaccine efficacy consistently in the last decade. 

### 2.6. A Machine Learning Model Predicting the Trajectories of Egg Passage Adaptation

From the analysis of the epistatic relationships between the 18 residues, we unraveled a fitness landscape with a rich epistatic network dominated by two negatively interacting residues, 186 and 194. Extensive epistasis leads to convergent adaptive walks in the fitness landscape during the egg passage adaptation. Since substitutions at residues 186 and 194 are often associated with a poor vaccine efficacy, the fitness landscape empowered by the epistatic network allows us to predict how a given candidate vaccine virus (CVV) will evolve in the direction of a certain evolutionary path, which might be dominated by residue 186 or 194 during the egg passage (Figure 3A). We thus constructed a computational tool: PEPA (Predicting Egg Passage Adaptation) that can predict the egg passage adaptation for any CVV before the passage experiment. The predictive model in PEPA can us help to select suitable CVVs with evolutionary trajectories unlikely to have mutations that negatively impact the vaccine efficacy (e.g., L194P). PEPA trains Random Forest or XGBoost on the labeled ancestral sequences before the egg passage adaptation and predicts whether the input sequence will have egg passage mutations at the two important residues, 186 or 194 (Figure 3B, Appendix A, Methods). By using a cross-validation design, PEPA achieved an overall accuracy of ~80% for predicting the residue changes in 186 and ~85% for predicting the residue 194 (Figure 3C). By comparison, we found that the XGBoost method predicts adaptive changes more accurately and robustly than the Random Forest model (Figure 3C). Based on its excellent performance, PEPA will be a powerful computational tool for guiding strain selection in future vaccine development studies.

## 3. Discussion

Using an unprecedent number of HA1 sequences from the GISAID database and a probabilistic approach known as mutational mapping, we found that egg passage adaptation has strong temporal fluctuations and is dominated by two major residues: 186 and 194 (Figure 1C). Substitutions in several important residues, including 186, 194, 160 and 225, are becoming increasingly frequent, suggesting that egg passage adaptation is growing progressively stronger. Analyzing the evolutionary trajectories of egg passage adaptation revealed an epistatic network, where residues 186 and 194 each form a sub-network with positive epistasis within each subnetwork but negative epistasis between subnetworks (Figure 2E). This epistatic network suggests that concerted changes across multiple residues drive egg passage adaptation. The temporal fluctuations in these residues were found to correlate with multiple cycles of rapid decline in vaccine efficacy (Appendix A). In addition, leveraging the highly convergent nature of egg passage adaptation, we built a machine learning model, PEPA, that can predict the evolutionary trajectories of any candidate vaccine virus before the egg passage experiment, providing a unique opportunity for the selection of optimal strains in future vaccine production. 

Egg passage adaptation provides one of the most extreme examples of adaptative evolution [47], with an estimated A(nonsyn)/S(syn) ratio of >30, a value rarely observed in any naturally evolving system [48]. This strong adaptation leads to repeated convergent or parallel evolutions, with specific mutations (e.g., G186V or L194P) contributing a majority of the changes involved in egg passage adaptation [35] and strong epistatic interactions between mutations in multiple residues. The reason for this extremely strong selection is likely rooted in the divergent cellular environments of avian and human cells. When influenza H3N2 originally crossed the species boundary in 1968, it was a bird-adapted virus with an initial capability of circulation in human populations, but since then, it has increasingly adapted to the human cellular and immunological environment [18,19]. As it evolves to increasingly adapt to the human host, it is becoming similarly maladapted to the avian environment, with increasing positive selection during egg-passaging. Unique combinations of amino acid substitutions (i.e., epistasis) facilitate the growth of embryonated eggs, leading to convergent patterns of evolutionary trajectories depending on the initial starting sequence. In addition, the number of significant codon positions has also increased from 12 to 18 in recent years [35]. As egg-passage adaptation becomes stronger over time, vaccine efficacy continuously declines. An additional factor that might contribute to this pattern is the fact that the circulating influenza strains may adapt to become maximally different at the sequence level from the optimally egg-adapted sequences, possibly due to selection imposed by the avoidance of vaccine-mediated immunity. In other words, circulating strains that differ maximally from the egg-adapted sequences are more likely to escape vaccine-induced immunity when vaccines are produced by egg passaging. 

Poor vaccine efficacy is often caused by a mismatch between the vaccine strains and the circulating strains [16]. As shown here, the strength of passage adaptation is highly predictive of the vaccine efficacy. A characterization of the fitness landscape of egg passage adaptation provides a unique means for guiding the choice of vaccine strains. The high predictability of egg passage adaptation allows us to select vaccine strains with evolutionary trajectories that are less likely to include the mutations that reduce vaccine efficacy during egg passage. For example, the G186V substitution, if occurring alone, increases the replication efficiency significantly in embryonated eggs but has a minimal impact on viral antigenicity [21,22,23,24]. Moreover, due to its negative epistasis with residue 194, substitutions in residue 186 can effectively block further substitutions in residue 194. Moreover, due to extensive epistatic relationships, one can imagine the benefits of editing specific mutations to create future vaccine strains to encourage the passage adaptation towards a path that is less likely to cause reduced vaccine efficacy. Thus, the fitness landscape can be employed to both predict and “design” the adaptive path for a better vaccine. 

There are a number of interesting directions worth pursuing in the future. First of all, even though many residues have frequent egg passage substitutions, the functional consequences of these substitutions have not been fully tested. For example, residue 225 is not present in any of the important functional domains (e.g., RBS or antigenic regions). We hypothesize that some of these substitutions might be compensatory mutations following mutations in other codon positions. Nevertheless, it will be important to experimentally test the functional consequences of these mutations and explore their links to vaccine efficacy in the future. Secondly, even for residues subject to existing functional experiments (e.g., testing the antigenic properties), their links to vaccine efficacy can still be further affected by their mutational frequencies in vaccine strains (e.g., residue 160, Appendix A), as well as the genetic background of the target strain. The study of temporal dynamics of egg passage adaptation provides an important platform for further refining important residues driving the low vaccine efficacy that has been observed over the years. Thirdly, we found that not only is the set of residues driving egg passage adaptation changing (e.g., from 12 to 18 residues), but the intensity of egg passage adaptation at different residues is also very dynamic. Considering the fact that vaccine production using embryonated eggs will likely continue for some time, it will be important for the field to develop new strategies for adapting to the dynamic landscape of egg passage adaptation. Lastly, in evolutionary biology, predicting a “future evolution” is quite difficult because of stochastic nature of evolution. The unique fitness landscape created by the highly epistatic network offers us a powerful machine learning model that can predict egg passage adaptation ahead of the passage experiment. In our work, the egg-passaged sequences constitute a minor proportion of the GISAID database. In other words, the model was only tested on a subset of influenza strains with egg-passaged sequences. It would be interesting to further test the accuracy of PEPA on more egg-passaged strains in the future. Moreover, as the predictive model only focused on the two dominant residues, it is important that we extend the model to predict substitutions in other codon sites (e.g., 160 or 225) in the future. 

Taken together, this study’s findings provide one of the most complete characterizations of the fitness landscape of any organism. They demonstrate how epistatic interactions drive viral evolution during egg passaging along predictable evolutionary paths in a multi-modal viral fitness landscape. Our results also show that a detailed understanding of this fitness landscape can be harnessed to design effective vaccines for seasonal influenza. 

## 4. Materials and Methods

### 4.1. Influenza Data Curation and Passage History Annotation

From the Global Initiative on Sharing All Influenza Data Epiflu (GISAID) database (https://www.gisaid.org/, 15 September 2020), we downloaded 96,745 hemagglutinin HA1 sequences for the H3N2 influenza and their associated passage annotations (Appendix A). In order to filter for viral strains of high quality, we removed sequences with (1) lengths that are too short (sequence length < 987), (2) unknown bases (not ATCG or degenerate bases), or (3) extremely long branches in the phylogenetic relationship (possibly due to low sequencing quality). After multiple QC steps, 89,853 sequences were retained for subsequent analysis (denoted as dataset D1). Due to extensive sequencing efforts over recent years, the GISAID database contains many identical sequences, and we thus created one non-redundant dataset consisting of only unique sequences (*n* = 37,938, denoted as dataset D2). 

Using the GISAID database, passage histories of the influenza sequences were collected from viral sequences submitted to the public database by individual researchers [44]. The passage histories were annotated using different syntaxes by individual researchers. For example, “E” and “EGG” can both indicate egg passage. In order to standardize the passage history, we used an approach similar to DuPai et al. [44] in order to curate passage histories for the entire dataset. In total, there were 9 passage categories, including Original (28,643), SIAT (15,288), RhMK (3002), Others (294), Mix (8463), Egg (989), NA (15,642), MDCK (6112) and Unknown Cell (11,420), for the “all sequences” dataset (D1) (Appendix A). Likewise, the corresponding passage histories in the data set of unique sequences (D2) included: Original (10546), SIAT (5173), RhMK (1332), Others (144), Mix (4444), Egg (716), NA (7082), MDCK (3127) and Unknown Cell (5599).

### 4.2. Sequence Alignment and the Phylogenetic Inference

We used MAFFT (Multiple Alignment using Fast Fourier Transform, version 7.464) [49] to perform the multiple sequence alignment of the two datasets (D1 and D2, with –auto option). IQ-TREE [45] (version 2.1.1) was used to infer the phylogenetic relationships of all the sequences with the GTR + I + Gamma model. Maximum likelihood estimates of the evolutionary parameters, including the phylogenetic tree and branch length estimates, evolutionary rates for each nucleotide position and the parameters of the GTR model, were used for the subsequent mutational mapping inference. The ETE Toolkit [50] was used for visualizing the phylogenetic tree (Figure 1B).

### 4.3. Inferring the History of Mutations Using Mutational Mapping

The mutational mapping method is a probabilistic procedure used to infer the history of mutations along the phylogeny for a given set of sequences [42,43,51]. The histories of mutations include both the ancestral states of all the internal nodes and the mutational changes along each branch. The inference procedure consists of three major steps: (1) calculating the conditional likelihood of the internal nodes recursively from the leaves of the tree up to the root, using the Felsenstein’s pruning algorithm [52], in which the conditional likelihoods at internal nodes are the joint probabilities of observing all descendants of the focal node of interest and the latent nucleotide states in the node; (2) sampling the state of the internal nodes recursively from the root of the tree down to the leaves; and (3) sampling the possible histories of the mutations conditioning on the states of the two nodes at the ends of each branch (edge). In this work, we employed the uniformization method [51], where we first sampled the number of changes along each branch and subsequently sampled the evolutionary path conditioning on the number of changes. With these three steps, we can efficiently sample possible evolutionary histories of the sequences according to their posterior probability [42,43]. 

### 4.4. Statistical Tests Identifying Residues Driving Egg Passage Adaptation

We developed three statistical tests to detect the residues driving egg passage adaptation. In the enrichment test, we asked whether mutations occurring at a specific residue tend to occur on the egg branches more often than expected by chance (Figure 1B). Let the total number of mutations in a specific residue across the whole tree be N and the number of mutations along the egg branches in this residue be n
*(*n≤N). Assuming that mutations arise independently on the tree, the probability of a random mutation occurring on the egg branches is $p_{e}=\frac{T_{e}}{T}$, where $\;T_{e}\;$ is the total length of the egg branches and T is the total tree length. The p-value for a specific residue is then calculated as the tail probability of observing at least n mutations using a binomial distribution $Bin\left(N,p_{e}\right)$:$$\mathrm{Pr}\left(X\geq n\right)=\overset{N}{\underset{i=n}{\sum }}\left(\begin{array}{c} N\\ i \end{array}\right)p_{e}^{i}{\left(1-p_{e}\right)}^{N-i}$$

In the positive selection test, we focused on mutations along the egg branches and examined whether the number of nonsynonymous substitutions (*N_N_*) was larger than expected when compared to the synonymous mutations (*N_S_*). For a given residue, the mutational history along the egg branches for a given sample history can be represented as a collection of codon statuses and their duration times. Specifically, we denoted all the sampled M codon statuses across the egg branches as {$S_{i}\in ℭ,\;i=1,2\ldots ,M$} and the corresponding duration times as $\left\{T_{i},\;i=1,2,\ldots ,M\right\}$, where ℭ is the set of sampled codon states along the egg branches and $T_{i}$ is the associated branch length at codon state $S_{i}$. For a given codon state *S_i_*, we define *p(S_i_)* as the conditional probability of a nonsynonymous change if a random mutation occurs in this codon, which can be computed as:$$p\left(S_{i}\right)=\frac{q_{N}\left(S_{i}\right)}{q_{N}\left(S_{i}\right)+q_{S}\left(S_{i}\right)}$$
where $q_{N}\left(S_{i}\right)$ and $q_{S}\left(S_{i}\right)$ are the total substitution rates to nonsynonymous and synonymous one-step neighboring codons of *S_i_*. To be more specific, for a given codon *S_i_*, there are *m* one-step nonsynonymous neighboring codons and *n* one-step synonymous neighboring codons. Thus, $q_{N}\left(S_{i}\right)$ and $q_{S}\left(S_{i}\right)$ can be computed as $q_{N}\left(S_{i}\right)=\overset{m}{\underset{j=1}{\sum }}q_{S_{i},S_{j}}$, $q_{S}\left(S_{i}\right)=\overset{n}{\underset{j=1}{\sum }}q_{S_{i},\;S_{j}}$, where $q_{S_{i},S_{j}}$ is the rate of the transition between codons with state *S_i_* and *S_j_* and can be retrieved from the transition matrices at the three nucleotide positions in the codon [53]. We computed the overall probability of nonsynonymous mutations along the egg branches for a given codon as a weighted average:$$p_{N}=\frac{\sum_{i=1}^{M}\left(p\left(S_{i}\right)\times T_{i}\right)}{\sum_{i=1}^{M}\left(T_{i}\right)}$$

Thus, the *p*-value for this specific residue was then calculated as the tail probability of observing at least $N_{N}$ nonsynonymous mutations using a binomial distribution:$$Bin\left(N_{N}+N_{S},\;p_{N}\right).$$

The convergence test examines whether mutations from a given codon status tend to occur repeatedly (convergently) with specific neighboring codons. For example, for a given codon with status C, there will be *I* one-step neighboring codons, denoted as:(1)$${ℭ}_{C}=\left\{C_{1},\ldots ,C_{I}:\forall i,\;C_{i}\;is\;onestep\;neighboring\;codon\;to\;C\;\right\}.$$

After obtaining a sample history from the mutational mapping of the egg branches, the observed number of mutations from C to its neighboring codon $C_{i}\in {ℭ}_{C}$ can be counted as $n_{C,i}$. Thus, the observed frequencies of the $C\to C_{i}$ mutation can be computed as $f_{C,i}=\frac{n_{C,i\;}}{\sum_{i}n_{C,i\;}},\;\forall i=1,\ldots ,I.$. In order to measure the level of convergence, we used the homogeneity score (*H*), defined as $H_{C}=\overset{I}{\underset{i=1}{\sum }}f_{C,i}^{2}$. The convergence test is conducted by examining whether the observed $H_{C}$ is larger than expected. 

In order to generate the empirical null distributions of the H scores, we performed random (unconditional) simulations according to the rate matrix inferred from the three nucleotide positions in the codon conditioning on the starting codon states for the given codon on the egg branches [42]. For any C, the H score calculated from the kth simulation is $H_{C,k}$. Thus, we obtained the empirical p-value for the observed $H_{C}$ as $p=\frac{\sum_{i=1}^{k}I_{(H_{C}\leq H_{C,i)}}}{k}$, where $I_{\left(A\right)}$ is the indicator of event A. In this study, we performed k=1000 simulations. 

For all statistical tests, multiple test corrections for all amino acid positions were performed using the false discovery rate (FDR) [54]. In order to account for uncertainties in the mutational mapping, we took an average of the FDR corrected *p*-values across multiple histories (*n* = 100). Amino acid positions with a mean q-value of < 0.05 in the positive selection test were inferred to be positions driving egg passage adaptation.

### 4.5. Temporal Dynamics of Egg Passage Adaptation across Residues

We defined the frequency of substitution for a given residue during a time interval as the mean number of substitutions in that residue per egg-passaged sequence within the given time interval. For example, let us assume that there is a set of n egg-passaged sequences in a given time window. After mutational mapping, for any sequence i in the set, there are $x_{ij}$ substitutions at codon position j. Then, the frequency of substitution of codon j in this time window is calculated as fj=∑i=1nxijn. Substitutions occurring on the internal branches within egg clades are counted in all their descendant sequences, as these substitutions are presumed to be convergently evolved and to have occurred independently of each other in the lineages leading to the leaf nodes. In order to estimate the rate of mutation of the egg passage, we took the estimated terminal branch lengths and divided them by the associated number of egg passages. 

### 4.6. An Epistatic Network of Residues Driving Egg Passage Adaptation

We used the contingency test to investigate whether pairs of residues tend to have substitutions along egg branches more or less frequently than expected if the mutations occur independently of each other (Figure 2B). In particular, for each pair of residues, we categorized all egg branches into B_00_ (no mutation in either residue), B_01_ (no mutation in the first residue but has mutations in the second residue), B_10_ (has mutations in the first residue but no mutation in the second residue) and B_11_ (has mutations in both residues). We categorized possible pairs of residues into positively (co-occurring) and negatively (mutually exclusive) interacting residues. The network structure reflecting these interacting relationship was drawn in R using igraph (https://igraph.org/, 15 September 2020), and the network modules are available at (https://www.rdocumentation.org/packages/dna/versions/2.1-2/topics/network.modules, accessed on 15 September 2020).

### 4.7. Enrichment Score (ES) 

The enrichment score (ES) [36] of a given allele (amino acid) at a particular amino acid position is defined as the P_egg_/P_all_, where P_egg_ is the frequency of the amino acid at the position in the egg strains and P_all_ is the allele frequency in all the known strain histories (i.e., excluding “Others”, “Mix”, “NA” and “Unknown Cell”). 

### 4.8. Vaccine Strains and Vaccine Efficacy (VE)

Vaccine strains for the H3N2 influenza virus developed in recent years (2010–2019) were retrieved from the Center for Disease Control and Prevention (CDC) website (https://www.fludb.org/brc/vaccineRecommend.spg?decorator=influenza, accessed on 15 September 2020). Egg-passaged sequences of the vaccine strains were retrieved from the GISAID database. Vaccine efficacy data were also retrieved from the CDC (https://www.cdc.gov/flu/vaccines-work/past-seasons-estimates.html, accessed on 15 September 2020) [8,9,10,11,12,13,14,15]

### 4.9. A Machine Learning Model for Predicting Egg Passage Adaptation

We developed a machine learning model in PEPA (Predicting Egg Passage Adaptation) to predict substitutions at the two dominant residues, 186 and 194, for a given candidate vaccine virus (CVV). We used the inferred ancestral sequences of egg-passaged sequences as the sequences for training the model before egg passage. According to the mutational histories inferred from the mutational mapping at a specific residue (e.g., 186 or 194), each ancestral sequence can be labelled as mutated or non-mutated. For multiple egg-passaged sequences within an egg clade, we randomly sampled one sequence, yielding a dataset of 688 sequences for the model. Since the proportion of mutated sequences is often small (i.e., unbalanced label), we applied an up-sampling procedure [55] to increase the number of sequences in the mutated category, thus producing a balanced dataset for the model training. We used Python package scikit-learn [56] to train the Random Forest and/or XGBoost models. We adopted a cross-validation procedure, which randomly selects 80% of the data for training and leaves 20% of the data for testing. The model performance was evaluated for its accuracy, precision, and recall, as well as the F1 score, calculated based on test sets from 100 replicate runs. The predictive model is available at https://github.com/LiuYunsongIOZ/Mutational_Mapping (accessed on 15 September 2020). 

## Figures and Tables

**Figure 1 viruses-14-02065-f001:**
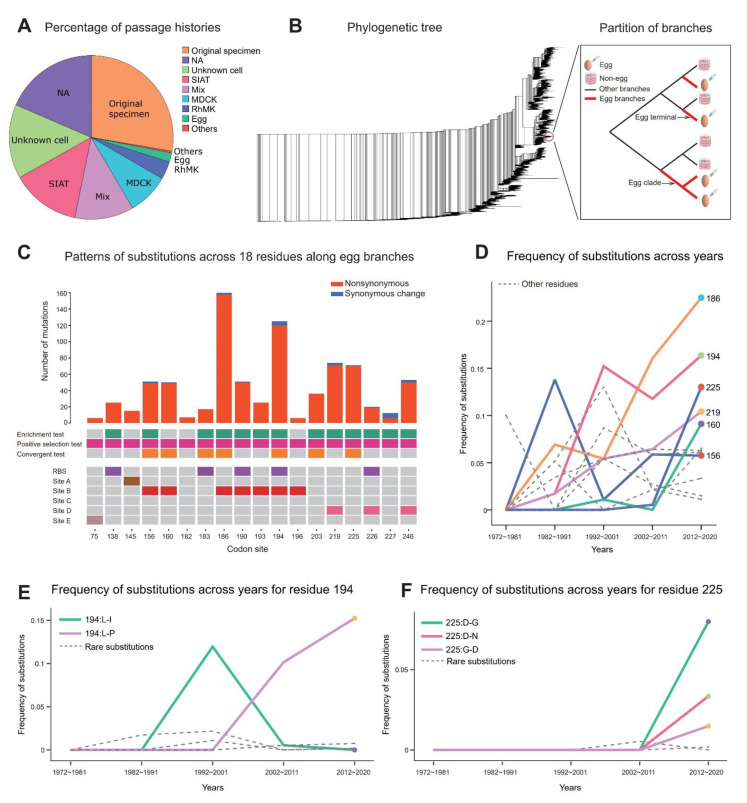
The temporal dynamics of egg passage adaptation. (**A**) The passage histories of the strains in the dataset (D1). (**B**) The phylogenetic relationships of all the sequences (Dataset D2). The inset illustrates the branches supporting egg passage adaptation (i.e., “egg branches”), which included both monophyletic groups for the egg-passaged sequences (i.e., “egg clade”), as well a single egg-passaged sequence (i.e., “egg terminal”). (**C**) The eighteen residues driving egg passage adaptation. The top bars indicate the number of nonsynonymous and synonymous changes (for a given sample history). The middle rows indicate the significance of the three statistical tests (Methods). The bottom plot shows whether the 18 residues are in the RBS (receptor binding sites) or antigenic domains A–E. (**D**) The temporal changes in the frequencies of substitutions at the 18 residues (only a few key residues, 156, 160, 219, 225, 194 and 186, are labelled). (**E**) The temporal changes in the frequency of different substitutions at residue 194. (**F**) The temporal changes in the frequency of different substitutions at residue 225.

**Figure 2 viruses-14-02065-f002:**
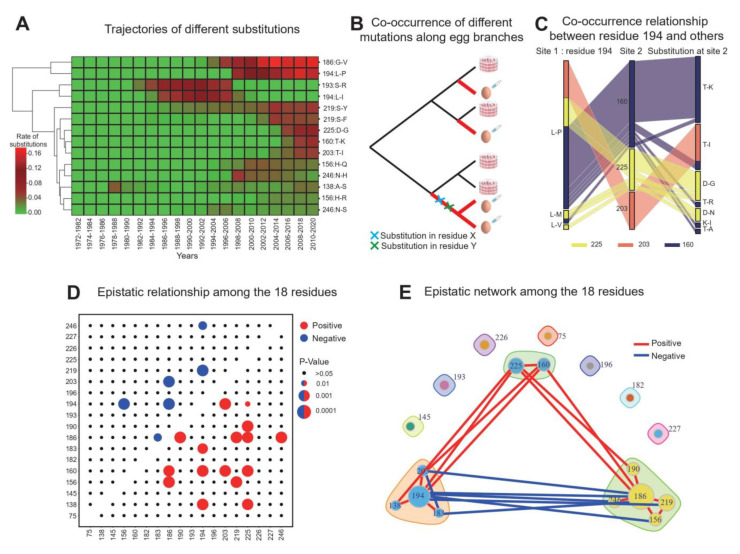
The epistatic landscape of egg passage adaptation. (**A**) Clustering temporal trajectories of the most frequent substitutions across the 18 residues (Methods). (**B**) Cartoon illustration of the tests of the co-occurrence relationship between two amino acid sites along the egg branches. (**C**) Examples of co-occurrence relationships between substitutions at different amino acid sites. The first column shows the substitution type at the first amino acid site, the second column indicates the second amino acid site, and the third column plots the types of substitutions occurring at the second amino acid site. Each line represents the history along one egg branch. In order to better illustrate different combinations of mutations, we colored the co-occurrence pattern based on different co-occurring amino acid positions (second column). Red: position 203; yellow: position 225; dark purple: position 160. (**D**) Positive and negative epistatic interactions between the 18 residues. Dot size indicates the level of significance. (**E**) The epistatic network between the 18 residues. Red and blue lines indicate positive and negative epistatic relationships. We have “clustered” the set of residues into discrete groups based on their epistatic properties. Residue 186 has a core group of positively epistatic residues, which tend to have negative epistatic relationship with the cluster centered around residue 194. The same pattern applies to the cluster around residue 194. Residues 160 and 225 tend to have positive epistatic relationships with two clusters surrounding 186 and 194.

**Figure 3 viruses-14-02065-f003:**
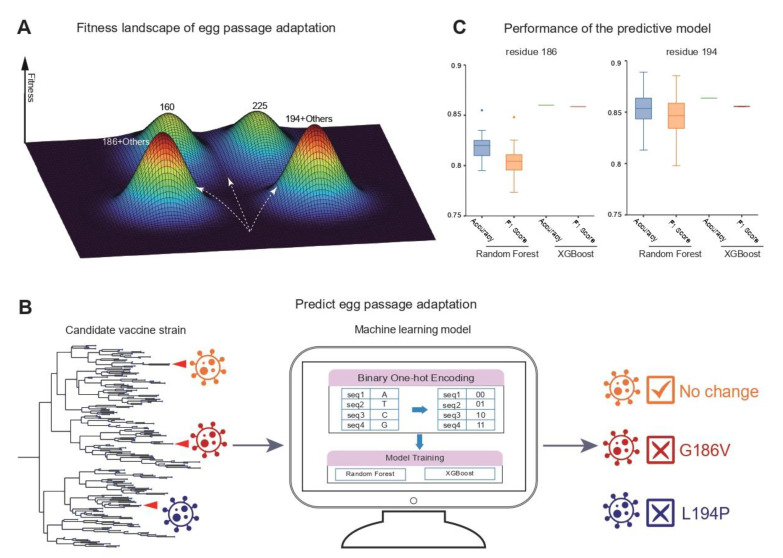
The fitness landscape and machine learning models. (**A**) A cartoon illustration of the fitness landscape of the egg passage adaptation. As residues 186 and 194 are two dominant codons correlated with vaccine efficacy, we thus summarized the evolutionary trajectories of egg passage adaptation into three possible evolutionary paths: (1) those with substitutions in residue 186 (no change in residue 194), (2) trajectories with substitutions in residue 194 (no change in residue 186) or (3) no changes in any of the two residues. (**B**) A schematic flow of the PEPA (Predicting Egg Passage Adaptation) (Methods). In the cross-validation, we trained the model on 80% of all the egg-passaged sequences (*n* = 688) and tested the model on the other 20% of the data. (**C**) The performance of the Random Forest and XGBoost model.

## Data Availability

Not applicable.

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
