# Peer review of "Predicting Egg Passage Adaptations to Design Better Vaccines for the H3N2 Influenza Virus"

_viruses, 2022, doi:10.3390/v14092065_

Round 1

Author Response

Reviewer 1:

The authors report on a correlation between some mutations in the HA of H3N2 influenza viruses acquired during egg passages and vaccine efficacy. While most residues corresponding to these mutations have been recorded, residues 160 and 225 were shown to increase in mutation frequency in the past two decades. Key residues 186 and 194 revealed mutually exclusive subnetworks, which consisted of positive epistatic interactions between some residues. Indeed, mutations in these two residues correlated with reduced vaccine efficacy. Further, the authors designed a machine learning tool (PEPA) that predicted with ~80% accuracy which H3N2 influenza viruses would acquire mutations in residue 186 or 194.

Reply: Thank you for the concise and precise summary of our work. We have addressed the detailed comments below.

General comments:

The authors provide a machine learning tool that can be used to prioritize some H3N2 influenza viruses for vaccine development, which is an advance in the field of influenza vaccine design. The model could be strengthened by introducing a test set of H3N2 influenza virus sequences with an unknown egg-passaged sequence and assessing the predictions by PEPA in ovo. This, however, may be out of the scope of the investigation and skill set of the authors. Some H3N2 strains may not have been passaged in eggs and thus lack an egg-adapted HA sequence on GISAID. In the absence of validating the model by in ovo passages, the authors would ideally provide additional discussion of this.

Reply: We want to thank the reviewer for the suggestion. Setting up passaging experiments growing different strains of influenza viruses is indeed beyond our specialty (a computational team). Even though we cannot setup extra experiments ourselves, in our statistical inference, we have implemented a cross-validation design by training the model in 80% of the data and predicting in the remaining 20% dataset. So, the predictive accuracy of our model has been calibrated based on existing passage experiments from the field (n=688). The reviewer is quite right that many of the strains in the GIDAID database was not passaged in eggs and will lack a corresponding egg sequence to “validate” our model.

In light of the reviewer’s suggestion, we have now acknowledged and discussed this point in the discussion (on page 11) of the latest version. “In our work, egg-passaged sequences contribute a minor proportion of the GISAID database. In other words, the model has only been tested in a subset of influenza strains with egg passaged sequences. It will be interesting to further test the accuracy of PEPA with more egg passaged strains in the future.”  

Minor comments:
While the term ‘codon’ was used to refer to the nucleotides encoding a given amino acid, it would be ideal to clarify this in the paper. Perhaps replacing ‘codon’ with ‘residue’ would be more ideal when referring to the amino acid number. In reference to the machine learning model, particularly the methods section, Figure 3, and Supplementary Figure 5, it would provide greater emphasis on the training set and test set of PEPA if the number of sequences for each set were provided. The introduction and methods sections have some grammatical errors, for example subject-verb agreement and the past participle of some verbs, that would ideally be addressed.

Reply: Thank you for pointing out several important points arising from the presentation which is quite helpful to us. Following the reviewer’s suggestions, we have: 1) We have replaced the usage of codon with residue in all places referring to the amino acid number. 2)We have added the number of sequences in the training and test dataset in the legend of Figure 3 and supplementary Figure 5. 3) We have now corrected the English presentation in our latest version.

We want to thank the reviewer for the constructive comments.

Reviewer 2 Report

Influenza virus has been regarding as the major pathogen leading to the next pandemics. Every year, trivalent flu vaccine is highly recommended to the young and the elderly in order to fight against the incoming dominant influenza virus. It is thus indispensable to have the best combination of virus strains in the vaccine. However, predicting that combination to fight against circulating strains is challenging, especially for the H3N2 subtype. Meanwhile, with the mass production of the flu vaccine in eggs, mutations on vaccine strains, resulted from the egg passage adaptation, are another issue to minimize the vaccine efficacy (VE). This study is the extension of the previous findings (Chen et al, 2019) from the same research team. In this study, Liu and colleagues analyze about 40,000 high quality and non-redundant H3N2’s hemagglutinin HA1 sequences since 1968. With a number of bioinformatic and statistical analysis, they identify frequently mutated 18 codons including codons 186 and 194 as reported in their previous study (Chen et al, 2019), and further determine their epistatic relationship and network. They also recapitulate the relationship between the VE and the codons 186 and 194 (Chen et al, 2019). Finally, based on this in-depth sequencing analysis, a machine learning model is built to predict the trajectories of egg passage adaption in order to select the H3N2 strains, not having the egg passage adaption, as the vaccine candidate. The machine learning model is appreciated, but number of major concerns must be addressed.

Major comments:

1.       As mentioned before, this study is the extension of the previous findings (Chen et al, 2019). In addition to the machine learning model, the novelty of other contents seems to be questionable.

2.       What is mutation rate of the virus when it is grown in egg for vaccine production?

3.       Two new codons, 160 and 225, are found, which should be regarded as the new findings. The codon 225 is located in neither RBS nor any antigenic sites. Why does it/ do they have high mutation rate? What is the implication? Do you have any functional studies of these codons? Meanwhile, these two codons are barely or unclearly discussed in terms of their biological importance or VE. What are the temporal changes in the frequency of substitutions at codon 160?

4.       Since the flu vaccine is mainly produced from eggs, egg passage adaption is an intrinsic problem for the vaccine preparation. As reported in the previous study (Chen et al, 2019), mutations at codons 186 and 194 must be screened out from the sequencing analysis now and even in the future. Why do you emphasize these two codons again in this study? On the other hand, it would be expected that more and more mutations induced by this adaption will occur, and the machine learning model will identify more and more this kind of mutation. In your previous, 12 frequently mutated codons are identified (Chen et al, 2019), while 18 codons are identified in this study. What is the implication? What is your comment? Are you suggesting that the flu vaccine production should be changed to other vaccine platform, like mRNA? Why does the substitution at codon 186 disappear after 2017? Instead of G186V, does it mutate to other amino acid?

5.       Since codon 186 or 194 are critical to the VE, with a simple sequence alignment, researchers are able to select the H3N2 strain without these two mutations as the vaccine candidate. What is the advantage of the machine learning model? The implication of the model is not clearly mentioned or discussed.  Why is codon 225 or 160 not taken into consideration in the machine learning model?

6.       It is confusing that there is no change at codons 160 and 225 in Fig 3A. More explanation is required.

7.       Where do you deposit the machine learning script or model?

Minor comments:

1.       Line number is missing in the manuscript. It is hard to specify errors in lines.

a.       Fig 2C, the color code is not clear, and the legend did not clearly describe the contents.

b.       Fig 2E, what does the circular area mean?

c.       ‘The temporal fluctuations in these codons were found to correlate with multiple cycles of rapid decline in vaccine efficacy.’ Any references?

d.       There is some typo- or grammatical error.

Author Response

Reviewer 2:

Comments and Suggestions for Authors

Influenza virus has been regarding as the major pathogen leading to the next pandemics. Every year, trivalent flu vaccine is highly recommended to the young and the elderly in order to fight against the incoming dominant influenza virus. It is thus indispensable to have the best combination of virus strains in the vaccine. However, predicting that combination to fight against circulating strains is challenging, especially for the H3N2 subtype. Meanwhile, with the mass production of the flu vaccine in eggs, mutations on vaccine strains, resulted from the egg passage adaptation, are another issue to minimize the vaccine efficacy (VE). This study is the extension of the previous findings (Chen et al, 2019) from the same research team. In this study, Liu and colleagues analyze about 40,000 high quality and non-redundant H3N2’s hemagglutinin HA1 sequences since 1968. With a number of bioinformatic and statistical analysis, they identify frequently mutated 18 codons including codons 186 and 194 as reported in their previous study (Chen et al, 2019), and further determine their epistatic relationship and network. They also recapitulate the relationship between the VE and the codons 186 and 194 (Chen et al, 2019). Finally, based on this in-depth sequencing analysis, a machine learning model is built to predict the trajectories of egg passage adaption in order to select the H3N2 strains, not having the egg passage adaption, as the vaccine candidate. The machine learning model is appreciated, but number of major concerns must be addressed.

Reply: Thank you for the precise and in-depth summary of our work.

Major comments:

Q1. As mentioned before, this study is the extension of the previous findings (Chen et al, 2019). In addition to the machine learning model, the novelty of other contents seems to be questionable.

Reply: Thank you for pointing out the gap in our presentation. There are a number of novel findings comparing to Chen et al 2019. 1) we have included more recent influenza H3N2 data which allow us to analyze much more sequences (i.e. 89,853 sequences in this study vs 32,278 sequences in Chen et al). With higher statistical power endowed by the bigger dataset, we were able to identify a much larger number of codon positions (18 in our study vs 12 from Chen et al). The frequencies of substitution at these residues have strong temporal fluctuations and many newly identified codon sites including 160 and 225 tend to have rapidly increasing frequencies of substitution during the past decade (Figure 1). 2) In this study, we were able to discover the epistatic landscape (network) of egg passage adaptation holistically (Figure 2). In other words, egg passage adaptation across multiple codon positions tends to mutate either concurrently (positive epistasis) or mutually exclusively (negative epistasis). This epistatic network has never been discovered before and is quite novel in the field of adaptative evolution. 3) Extending the initial finding in Chen et al, we found that the strength of egg passage adaptation is consistently correlated with the poor vaccine efficacy in recent years. This suggests that egg passage adaptation is not a sporadic factor, but rather a consistent force driving low vaccine efficacy. 4) Integrating the findings in 1/2/3, we were able to construct a novel machine learning model implemented in PEPA to select the best candidate vaccine strains that are less likely to accumulate deleterious mutations that will affect the vaccine efficacy.

In light of the reviewer’s comment, we have re-emphasized the novelty of this work in the first paragraph of discussion (page 10) as:

“Using an unprecedent amount of HA1 sequences from the GISAID database and a probabilistic approach known as mutational mapping, we found that egg passage adaptation has strong temporal fluctuations and is dominated by two major residues: 186 and 194 (Fig. 1C). Substitutions in several important residues including 186, 194,160 and 225 are becoming increasingly more frequent, suggesting that egg passage adaptation is getting progressively stronger. Analyzing evolutionary trajectories of egg passage adaptation revealed an epistatic network where residue 186 and 194 each form a sub-network with positive epistasis within each subnetwork, but negative epistasis between subnetworks (Fig. 2E). This epistatic network suggests that concerted changes across multiple residues drive egg passage adaptation. The temporal fluctuations in these residues were found to correlate with multiple cycles of rapid decline in vaccine efficacy (Supplementary Fig. 4). In addition, leveraging the highly convergent nature of egg passage adaptation, we built a machine learning model, PEPA, that can predict evolutionary trajectories for any candidate vaccine virus before the egg passage experiment, providing a unique opportunity for selecting optimal strains for future vaccine production.”

Q2. What is mutation rate of the virus when it is grown in egg for vaccine production?

Reply: This is a very good and also a challenging question. Accurately measuring the mutation rate per “generation” for viruses is actually quite difficult because the number of generations viruses grow in the passage medium is difficult to measure accurately (e.g. asynchronized infection cycles). However, we can get a rough estimate of the mutation rate per passage. We first took the (estimated) terminal branch lengths leading to egg-passaged sequences. In order to estimate the mutation rate per passage, we divided the branch lengths by their associated number of passages growing the viral strain in embryonated eggs, we estimated the mutation rate to be 0.00144 per site per passage.

Q3. Two new codons, 160 and 225, are found, which should be regarded as the new findings. The codon 225 is located in neither RBS nor any antigenic sites. Why does it/ do they have high mutation rate? What is the implication? Do you have any functional studies of these codons? Meanwhile, these two codons are barely or unclearly discussed in terms of their biological importance or VE. What are the temporal changes in the frequency of substitutions at codon 160?

Reply: There are several questions raised in this point. We would like to address them one by one. 1) The reviewer is quite right that codon 160 and 225 are two newly identified codons with increasing rates of egg-passage mutations in recent years. 2) It is also quite interesting to us that codon 225 is not in RBS or antigen sites. The reason for its high mutation rate is not yet fully clear for now. We hypothesis that these mutations in codon 225 might be compensatory mutations complementary (i.e. epistatic) to substitutions in major codons (e.g. 186 or 194). 3) From the literature survey, substitutions at codon position 160 were found to be associated with vaccine efficacy (e.g. PMID 29109276). The experimental data for codon 225 is very rare, possibly due to its recent activity. 4) From Figure 1D, we found that substitutions at codon position 160 are also getting progressively more frequent.

In light of the reviewer’s comments, we have now discussed these points in the discussion (page 11) as: “Even though many residues have frequent egg passage substitutions, functional consequences of these substitutions have not been fully tested. For example, residue 225 are not in any of the important functional domains (e.g. RBS or antigenic regions). We hypothesize that some of these substitutions might be compensatory mutations following mutations in other codon positions. Nevertheless, it will be important to experimentally test functional consequences of these mutations and explore their link to vaccine efficacy in the future.”

Q4. Since the flu vaccine is mainly produced from eggs, egg passage adaption is an intrinsic problem for the vaccine preparation. As reported in the previous study (Chen et al, 2019), mutations at codons 186 and 194 must be screened out from the sequencing analysis now and even in the future. Why do you emphasize these two codons again in this study? On the other hand, it would be expected that more and more mutations induced by this adaption will occur, and the machine learning model will identify more and more this kind of mutation. In your previous, 12 frequently mutated codons are identified (Chen et al, 2019), while 18 codons are identified in this study. What is the implication? What is your comment? Are you suggesting that the flu vaccine production should be changed to other vaccine platform, like mRNA? Why does the substitution at codon 186 disappear after 2017? Instead of G186V, does it mutate to other amino acid?

Reply: Thanks for the thoughtful questions. We would like to address them one by one. 1) The mutational mapping method allowed us to study the overall pattern of substitutions across many repeated passage experiments. One of the major findings from this study is the epistatic relationship among multiple residues. We found that codon position 186 and 194 are not only the two most frequently mutated residues (Fig 1C), but also dominate the first and second principal components (PCs) in the adaptive distance (Supplementary Fig. 3B). In addition, these two residues are not only mutually exclusive with each other (Fig. 2D), but also had strong epistatic relationship with multiple other codons (Fig. 2E). This epistatic network centered around the two dominant codons (186 and 194) has not been discovered and is quite novel. 2) The reviewer is quite right that the number of codons driving egg passage adaptation is increasing through the years (i.e. from 12 residues to 18 residues) with time varying intensities (Figure 1). This suggests that egg passage adaptation is a very dynamic progress through time. These observations suggest that it is important for the field to routinely trace the temporal dynamics in egg passage adaptation. 3) We agree with the reviewer that mRNA vaccines can be one of the ideal solutions out of this problem. However, due to the tight regulation and low profit margin for influenza vaccines, producing vaccines in egg medium will likely continue for some time. 4) The reviewer is quite right that there is no substitution at codon position 186 in vaccine strains after 2017 (but substitutions at 186 are very frequent in egg passaged sequences in general, Figure 1). The reason they are not appearing in vaccine strains is due to the negative epistatic interaction with codon 194 as recent vaccine strains tend to have egg passage adaptation in 194.

In light of the reviewer’s comments. We have added these points to the discussion of the article (page 11) as: “We found that not only the set of residues driving egg passage adaptation is changing (e.g. from 12 to 18 residues), but the intensity of egg passage adaptation at different residues are also very dynamic. Considering the fact that vaccine production with embryonated eggs will likely continue for quite some time, it will be important for the field to develop new strategies adapting to the dynamic landscape in egg passage adaptation.”

Q5. Since codon 186 or 194 are critical to the VE, with a simple sequence alignment, researchers are able to select the H3N2 strain without these two mutations as the vaccine candidate. What is the advantage of the machine learning model? The implication of the model is not clearly mentioned or discussed. Why is codon 225 or 160 not taken into consideration in the machine learning model?

Reply: 1) The reviewer is quite right that if we can perform post-passage sequencing, we can filter out strains with mutations in 186 or 194. The power of the machine learning model is that we can predict whether codon 186 or 194 will have passage substitution before the egg passage experiments. This can be very important in the vaccine production as it is not easy to “change the strains” after the WHO meeting. In other words, it is a powerful computational tool predicting egg passage adaptation ahead of the actual production. 2) Since the vaccine efficacy is mainly associated with the two dominant codon 186 and 194, we have only predicted passage adaptation in codon 186 and 194 (not yet for 225 or 160). We will pursue an extended model incorporating other codons in a future study.

In light of the reviewer’s comment, we have now re-emphasized the potential utility of the machine learning model in the maintext. Also, we also added new materials to the discussion (page 11) as “In evolutionary biology, predicting ‘future evolution’ is quite difficult because of stochastic nature in evolution. The unique fitness landscape driven by the highly epistatic network endowed us a powerful machine learning model that can predict egg passage adaptation ahead of the passage experiment….Moreover, the predictive model has only focused on the two dominant residues, it will be important to extend the model to predict substitutions in other codon sites (e.g. 160 or 225).”

Q6. It is confusing that there is no change at codons 160 and 225 in Fig 3A. More explanation is required.

Reply: Figure 3A is a cartoon model illustrating the fitness landscape. Based on the substitution pattern in the two dominant residues 186 and 194, we have categorized the passage adaptation into three evolutionary trajectories. Because of the strong negative epistasis between 186 and 194, we have categorized the evolutionary trajectories into 1) trajectories with substitution in residue 186, 2) trajectories with substitution in 194, 3) trajectories with substitutions in neither of the two residues. Figure 3A is meant to illustrate there are three major evolutionary trajectories in egg passage adaptation. We have now added these information to the latest figure legend.

Q7. Where do you deposit the machine learning script or model?

Reply: we have now deposited the script at

https://github.com/LiuYunsongIOZ/Mutational_Mapping

Minor comments:

  1. Line number is missing in the manuscript. It is hard to specify errors in lines.

Reply: Sorry for the inconvenience. We had line numbers added in the preview version. Somehow, it is not presented in the version given to the reviewer (Maybe a different version of the main-text was given, we are not very sure).  

  1. Fig 2C, the color code is not clear, and the legend did not clearly describe the contents.

Reply: Thank you for pointing out the gap in the presentation. The color was used to separate different types of changes. It was used for illustration purposes. We have added the explanation to the figure legend.

  1. Fig 2E, what does the circular area mean?

Reply: Thank you for pointing out the gap in the presentation. Different circles represent clusters of residues with similar epistatic properties. We have now added the explanation to the figure legend.

  1. ‘The temporal fluctuations in these codons were found to correlate with multiple cycles of rapid decline in vaccine efficacy.’ Any references?

Reply: we have now added new references.

  1. There is some typo- or grammatical error.

Reply: we have now corrected our English.

We want to thank the reviewer for the constructive and helpful comments.

Round 2

Reviewer 2 Report

The revised version is highly improved, and the importance and novelty of the study is clearly presented now. Some concerns have to be addressed further.

A.    Some responses from the cover letter are highly recommended to be included in the manuscript.

1.    The mutation rate of the virus, grown in eggs for vaccine production, should be included in the discussion or main text, because that estimated mutation rate (0.00144/site/passage) should be supportive for the implication of the machine learning model.

2.    The webpage where the authors deposited the script.

B.    In the previous review, the temporal changes in the frequency of substitutions at codon 160 was asked, and the authors tried hard to address this question. The main reason to raise the above question is that the authors did not clearly explain why the frequency of substitutions across codon 225 is focused but the related analysis of codon 160 is excluded. Given that codon 160 is associated with vaccine efficacy instead of codon 225.

C.   In the first column of Fig 2C, there is no label for the red, yellow and the small dark purple bars. Authors have to explain the absence of the label. Does it mean that there is no change at the amino acid?

Author Response

Reviewer 2:

Comments and Suggestions for Authors

The revised version is highly improved, and the importance and novelty of the study is clearly presented now. Some concerns have to be addressed further.

Reply: Thank you for the constructive suggestions in the previous review and the encouraging comments here.

A. Some responses from the cover letter are highly recommended to be included in the manuscript. 1. The mutation rate of the virus, grown in eggs for vaccine production, should be included in the discussion or main text, because that estimated mutation rate (0.00144/site/passage) should be supportive for the implication of the machine learning model.2. The webpage where the authors deposited the script.

Reply: We have now added the mutation rate to the subsection “Strong temporal fluctuations in egg passage adaptation” of the manuscript and the webpage to the M&M of the latest manuscript.

B. In the previous review, the temporal changes in the frequency of substitutions at codon 160 was asked, and the authors tried hard to address this question. The main reason to raise the above question is that the authors did not clearly explain why the frequency of substitutions across codon 225 is focused but the related analysis of codon 160 is excluded. Given that codon 160 is associated with vaccine efficacy instead of codon 225.

Reply: The reviewer pointed out an important point. In the previous review, the reviewer asked us to discuss the relationship between residue 160/225 and vaccine efficacy, in particular, it is quite surprising as residue 225 is not in any of the functional domains. In the previous revision, we have discussed potential factors (e.g. compensatory mutations) that might drive the high frequency of mutations at residue 160 or 225. We didn’t further discuss the functional role of residue 160. Even though residue 160 frequently mutates in recent years (e.g. T->K change), substitutions in vaccine strains are rather rare (Supplementary Figure 4). At this point, it is still unclear how residue 160 might affect vaccine efficacy across the years. We have chosen to stay conservative in the previous revision given the above background.

In light of the reviewer’s comment, we have further discussed this point in the fourth paragraph of Discussion section as “even for residues with existing functional experiments (e.g. testing antigenic properties), their link to vaccine efficacy can still be further affected by their mutational frequencies in vaccine strains (e.g. residue 160, Supplement Fig. 4) as well as the genetic background of the target strain. The study of temporal dynamics of egg passage adaptation provides an important platform further refining important residues driving low vaccine efficacy across years”

C. In the first column of Fig 2C, there is no label for the red, yellow and the small dark purple bars. Authors have to explain the absence of the label. Does it mean that there is no change at the amino acid?

Reply:Thank you for pointing out the gap in the presentation. Figure 2C was drawn to show the epistatic relationship between residue 194 and others. Because there are multiple combinations of sites and mutations, we have plotted the changes in residues 194 (first column), the co-mutated sites (in the second column) and substitutions in the co-mutated site (third column). Each line represents one egg-passage experiment. In order to better illustrate different combinations of mutations, we colored the co-occurrence pattern based on different co-occurring amino acid positions (second column). Different colors were used for the illustration purpose.

We have now added the following interpretation and explicitly indicated the color used in the figure to the legend of Figure 2 as: “In order to better illustrate different combinations of mutations, we colored the co-occurrence pattern based on different co-occurring amino acid positions (second column). Red: position 203, Yellow: position 225, Dark purple: position 160.”
